# Residue Levels and Dietary Intake Risk Assessments of 139 Pesticides in Agricultural Produce Using the m-PFC Method Based on SBA-15-C_18_ with GC-MS/MS

**DOI:** 10.3390/molecules28062480

**Published:** 2023-03-08

**Authors:** Yue Wang, Tingjie Huang, Tao Zhang, Xiaoping Ma, Guangshuo Zhou, Meiyao Chi, Xinjie Geng, Chunhao Yuan, Nan Zou

**Affiliations:** 1School of Chemistry and Pharmaceutical Engineering, Shandong First Medical University, Shandong Academy of Medical Sciences, Tai’an 271016, China; 2Key Laboratory of Pesticide Toxicology & Application Technique, College of Plant Protection, Shandong Agricultural University, Tai’an 271018, China

**Keywords:** multiplug filtration cleanup, SBA-15-C_18_, pesticide residue, dietary risk assessment

## Abstract

A survey was designed to investigate the pesticide residues in agricultural produce and to estimate their potential intake risks to inhabitants. A total of 314 samples of nine types of fruits and vegetables were collected from the supermarkets and vegetable markets of Shandong Province (China) from October 2020 to February 2022. An accurate and reliable multi-residue method, based on GC-MS/MS detection, as well as the multiplug filtration cleanup method, based on SBA-15-C_18_, was prepared by a solution chemical reaction. Additionally, an in situ co-condensation method was established for the quantification of 139 pesticide residues. Residues that contained no pesticides were detected in 66.5% of the 314 samples. Moreover, of the samples, 30.6% were at or below the MRLs, and 2.9% were above the MRLs. Residues of procymidone were found to be the one that most often exceeded the MRLs (1.3% of the samples). Tebuconazole was found most frequently in 22.0% of the samples analyzed. Consumer exposure to the 139 pesticides did not exceed 100% ADI and ARfD. This led to a consideration that these pesticide residues in the nine commodities may not raise the health risk of the consumers in the long and short term. The highest value of chronic dietary intake was obtained from spirodiclofen, which resulted in a 24.1% of ADI. Furthermore, the highest exposure levels in the short term were obtained from the consumption of leeks with procymidone (58.3% ARfD).

## 1. Introduction

Pesticides are widely used in fields during crop production to control diseases, pests, and weeds to ensure yields and the quality of agricultural crops. Despite the obvious benefits of the use of pesticides, there has been a growing concern about pesticide residues in the environment and to food [1]. In addition, the increased use of chemical pesticides has caused many associated short-term or long-term effects on human health [2]. Pesticides have been associated with a range of human health problems, ranging from acute effects, such as headaches and nausea, to chronic impacts, such as cancer, reproductive harm, and endocrine disruption [3].

Fruits and vegetables have received an increasing amount of attention in monitoring programs since most of them are eaten raw. Further, they may contain higher pesticide residue levels compared to other food groups [4,5,6]. Moreover, the human intake of toxic substances due to pesticide residues in food may be much higher than the intake of these substances through the acts of drinking water and inhaling air [7]. Therefore, it is very important to monitor pesticide residues in fruits and vegetables, as well as to assess whether they pose a risk to human health [8]. Many countries have set up pesticide residue monitoring systems and there have been a number of reports on the pesticide residues detected in crops [9,10], fruits [11,12], vegetables [11,13,14], medicinal plants [15], milk, [16] and fish [17]. The monitoring has focused on the rational use of pesticides in areas of authorization and registration (pesticide application and harvest intervals), as well as in compliance with maximum residue limits (MRLs). Intake risk assessments of pesticide residue were also recorded in many countries and regions [8,11,15,17]. Shandong Province is the largest vegetable planting province and vegetable export province in China. The Shouguang Vegetable Wholesale Market in Shandong Province supplies vegetables to more than 20 provinces [18]. A case study in Anqiu City estimated that more than 50% of the vegetables produced were exported, 30% were sold to the domestic market, and 20% were sold to the local market [19]. Therefore, the monitoring and risk assessment of pesticide residues within the typical fruits and vegetables in Shandong Province is of great significance.

Multi-walled carbon nanotubes (MWCNTs) are interesting carbonaceous materials with an ultra-high specific surface area. They have been applied, in the pesticide multi-residue analysis that was conducted in our previous work, as effective cleanup sorbents for the removal of fatty acids, pigments, and other impurities of medium and high polarity [20,21,22]. The mesoporous material SBA-15 and its derivatives are characterized by high adsorption versatility, strong adsorption stability, and large adsorption capacity. They are widely used to separate and remove organic pollutants and heavy metals in food and environments [23,24,25]. In this study, SBA-15-C_18_ was prepared by introducing C_18_ groups into the active site of SBA-15, which exhibited better adsorption properties for non-polar and weakly polar compounds compared to SBA-15 [26,27]. SBA-15-C_18_ showed the characteristics of a high specific surface area, high pore volume, and uniform pore distribution, as well as had higher recovery values than were found in commercial C_18_ amorphous silica [28]. Our research group has developed a fast, simple, and effective purification process called the multi-plug filtration cleaning method (m-PFC), which has been widely used in the detection of pesticide residues in agricultural products [21,22,29]. Through the use of m-PFC, the adsorbent mixture was fixed by two layers of sieve plates and was then filled into the syringe tube. The m-PFC process uses a streamlined filtration process with a very fast purification rate of just a few seconds, eliminating the need for whirlpool, centrifugation, and solvent evaporation steps.

The aim of the present study is to develop an m-PFC method based on MWCNTs and SBA-15-C_18_ with GC-MS/MS for the multi-residue analysis of 139 pesticides. Furthermore, in this study, the presence of the selected pesticides that are commonly used on 9 agricultural products is investigated; further, their compliance with the current maximum limit standards are also assessed. Combined with the results of monitoring projects and food consumption data, the dietary intake risk of pesticide residues in agricultural consumption was assessed in order to establish practical guidelines for monitoring the use of pesticides in the agricultural industry. The results can be used to design future control precepts in the region and to take preventive measures to minimize risks to human health.

## 2. Results and Discussion

### 2.1. Characterization of SBA-15-C_18_

The FTIR spectra (Figure 1B,C) of the synthesized SBA-15-C_18_ and C_18_ showed stretching vibration at 2926 and 2855 cm^−1^. This was also the stretching vibration of CH_2_ in the C_18_ group, indicating that the C_18_ group was successfully introduced and that SBA-15-C_18_ was successfully synthesized. The SEM figures (Figure 1D,E) showed that SBA-15-C_18_ had two-dimensional through pore and regular pore structures, and the particle sizes were uniform, cylindrical, and curved with an average particle size between 240 and 340 nm.

### 2.2. Determination of Pesticide Residues

The mixtures of MWCNTs, PSA, and C_18_ have been shown to have excellent ability in selectively removing interfering substances from the acetonitrile extracts of the troublesome leek matrices [22]. SBA-15-C18 could achieve a better purification effect with less dosage compared with C18. For example, for leek samples, the purification materials in the published literature were 8 mg MWCNTs, 10 mg PSA, 10 mg C18, and 150 mg MgSO_4_ [22], which were 8 mg of MWCNTs+, 10 mg of PSA+, 8 mg of SBA-15-C18+, and 150 mg of MgSO_4_ in this study. For the other matrices, the purification effect of the different combinations and proportions of purification materials was conducted on the spiked extracts. The final sorbent combination and proportion for the different matrices were as per the following: 5 mg of MWCNTs, 5 mg of SBA-15-C_18_, and 150 mg of MgSO_4_ for the watermelon, melon, asparagus, lotus root, strawberry, and cucumber samples; 8 mg of MWCNTs, 10 mg of PSA, 8 mg of SBA-15-C_18_, and 150 mg of MgSO_4_ for the leek samples; and 8 mg of MWCNTs, 5 mg of SBA-15-C_18_, and 150 mg of MgSO_4_ for the crown daisy and leaf lettuce samples.

Matrix-matched standard calibrations for each matrix were used for the more accurate results in order to avoid the matrix effect. The standard curves of all pesticides were in the range of 10–500 μg L^−1^ by the calculation of a five-point plot (10, 50, 100, 200, and 500 μg L^−1^), and a good regression correlation (R^2^ > 0.99) was achieved for all pesticides (the relevant results of the correlation equation and R^2^ are presented in Appendix A (Appendix A) with leek samples taken as an example). The LOQs of the proposed method were determined by the lowest accepted fortification level, which was analyzed by the Xcalibur Data System. The LOQs ranged from 1 to 10 µg kg^−1^, and the LOQs of each compound were detailed in Appendix A. Average recoveries were conducted at one fortification level (10 μg kg^−1^) for all the pesticides in the nine types of fruits and vegetables, and were in the range from 71.1 to 120.6% with the relative standard deviations below 16.5% (Appendix A). Therefore, through the validation results of accuracy, precision, and sensitivity, the method met the requirements for the detection of multiple pesticide residues in the nine kinds of agricultural products.

### 2.3. Pesticide Residues in Samples

#### 2.3.1. Evaluation by Samples

The overview of the data obtained after the analysis of the 314 samples is shown in Figure 2. A total of 209 samples (66.5%) showed that no pesticide residues were detected. A total of 96 (30.6%) analyzed samples contained pesticide residues at or below the MRLs of China (GB 2763-2021). In addition, 9 samples (2.9%) contained pesticide residues above the MRLs. The order of percentages of pesticides above the MRLs was as follows: leek (7.1%) > leaf lettuce (5.6%) > cucumber (4.8%) strawberry (3.1%) > melon (2.9%). No pesticide residues in the watermelon and leaf lettuce products exceeded the MRLs.

#### 2.3.2. Evaluation by Pesticides

In this monitoring program, 4 pesticide residues were found to exceed MRLs in 9 samples, which were triadimenol in 1 melon sample and 1 cucumber sample; procymidone in 3 leek samples and 1 cucumber sample; chlorpyrifos in 1 leaf lettuce sample; and cypermethrin in 1 strawberry sample and 1 cucumber sample. The residues of procymidone were found to be the ones most often exceeding the Chinese MRLs (1.3% of the samples). Table 1 shows the specific detected amounts of several pesticides exceeding the MRL. The order of percentage of pesticides exceeding the MRLs was as follows: procymidone (1.3%) > cypermethrin (0.6%) = triadimenol (0.6%) > chlorpyrifos (0.3%). The mean levels of residues and concentration ranges of 139 pesticide in 9 agricultural products are presented in Appendix A. Procymidone is a fungicide used as a seed dressing, pre-harvest spray, or post-harvest dip for the control of various fungal diseases, and it is likely to be commonly used in many countries.

Residues were found most frequently of tebuconazole (22.0%), followed by procymidone (20.4%), chlorpyrifos (18.5%), boscalid (17.2%), triadimenol (16.2%), pyriproxyfen (11.5%), butachlor (9.9%), cyfluthrin (9.2%), fenpropathrin (8.6%), etc. Tebuconazole was detected in the concentration range of lower than the detection limit (DL) to 1.599 mg kg^−1^, most were from the leek and crown daisy samples. As the pesticide with the highest detection rate, tebuconazole is an efficient and broad-spectrum triazole bactericide with a good internal absorption. This pesticide has good efficacy and wide application range in China, so the detection frequency is relatively high, but it does not exceed the limit standard [30].

The detected 139 pesticides obtained for the mean, maximum, and minimum values (mg kg^−1^) are listed in Appendix A. The mean concentrations of pesticides in the total samples were the highest for chlorfenapyr (2.853 mg kg^−1^). The maximum concentrations of pesticides found in the total samples were the highest for procymidone (11.27 mg kg^−1^). In this study, the procymidone in the three leek samples exceeded the MRL. In fact, the phenomenon of procymidone exceeding the MRL in leek often occurs [31,32]. This is related to the application method of procymidone with a smoke generator that is commonly used in China. The MRL of procymidone in Chinese leeks is 0.2 mg/kg, which refers to the standard of CAC. According to the actual production in China, the residue of procymidone in leeks is seriously excessive, resulting in procymidone not being registered in leeks. To solve the problem, relevant departments of the Ministry of Agriculture of China have started to revise the standards.

#### 2.3.3. The Results of Multiple Pesticide Residues

Figure 3 shows the commodities that presented multiple residues. The residues of two or more pesticides were found in 87 (75.2%) of the positive samples. Cucumber was the agricultural product with the highest number of samples with multiple residues (85.7% of the positive cucumber samples), followed by leek (83.3%), and then by watermelon (81.8%). With the exception of asparagus and lotus root, all samples had mainly two or three residues present. Two leek samples had six residues, one had eight residues, and one strawberry sample had seven different residues.

### 2.4. Risk Assessment of the Long-Term Intake and Chronic Exposure

The chronic intake risk was evaluated by Equation (1). The chronic intake risk assessment of the monitored agricultural product was based on the reported calculations, which are presented in Table 2. The chronic intakes of all the selected pesticide residues considered were rather low compared to the ADI, indicating that the pesticide residues in the agricultural products were within the acceptable range. The highest value of chronic exposure was obtained from spirodiclofen, which resulted in 24.1% of the ADI. Consequently, the safety of the Shandong Province consumer generally seems to be under control in terms of the pesticides that are consumed through the long-term consumption of the monitored fruits and vegetables.

### 2.5. Risk Assessment of the Short-Term Intake and Acute Exposure

The results for the short-term exposure risk assessment regarding the intake of the monitored fruits and vegetables were calculated, as shown in Table 3. The consumer acute exposure to pesticides does not exceed the value of 100% ARfD. The highest values of short-term exposure were obtained in the case of the consumption of leeks with procymidone (58.3% ARfD), followed by leeks with chlorfenapyr (49.2% ARfD), and then leaf lettuce with procymidone (46.8% ARfD). In the remaining cases, the ARfD% values from the other pesticides were between 0.0 and 21.2%, indicating that the acute exposure risk in within an acceptable range. The residue level could not be considered a serious public health problem in the short-term exposure. The only noted possible risk was connected with procymidone residues in the leek and leaf lettuce.

## 3. Materials and Methods

### 3.1. Standards, Reagents and Materials

The pesticide standards, with a purity of 95~99%, used in this work (Appendix A) were provided by the Institute of the Control of Agrochemicals, Ministry of Agriculture (Beijing, China). The working standard mixture containing 10 mg L^−1^ of each pesticide was formulated with acetone and stored at −20 °C. The PSA and C_18_ were purchased from Tianjin Bonna-Agela Technologies Co., Ltd. (Tianjin, China). The MWCNTs, with average diameters of 5–10 nm, were purchased from the National Center for Nanoscience and Technology. The P123, TEOS, and octadecyldimethylchlorosilane were purchased from Sinopharm Chemical Reagent Co., Ltd., (Shanghai, China). The HPLC grade acetonitrile and acetone were obtained from Fisher Chemicals (Fair Lawn, NJ, USA). The NaCl and MgSO_4_ were purchased from Sinopharm Chemical Reagent (Beijing, China).

### 3.2. Sample Collection

A total of 314 agricultural product samples were collected from October 2020 to February 2022. The samples were collected from randomly selected supermarkets, markets, and vegetable bases from each of the 10 cities of Shandong province. All samples were obtained from local production, and the specific information of samples are shown in Appendix A. The sampling was carried out with the assistance of authorized personnel from the food control authorities in the districts involved. The fresh agricultural product samples analyzed in this study included watermelon, melon, asparagus, lotus root, strawberry, leek, crown daisy, leaf lettuce, and cucumber.

The sampling was conducted in accordance with the guidelines in China (SAC, 2014) for the official control of pesticide residues. All samples were comminuted after being transported to the laboratory and stored at −20 °C until analysis was carried out.

### 3.3. GC-MS/MS Analytical Conditions

The analysis was carried out on a Thermo Scientific^TM^ TSQ^TM^ 8000 Evo triple quadrupole mass spectrometer. Samples were injected with an AI 1310 auto-sampler into a split/splitless injector. The capillary column was analyzed with a Thermo Fisher Scientific TR-5MS (30 m × 250 μm × 0.25 μm).

At 0.75 min, the split mode was switched, and the split flow was 60 mL min^−1^. At 2 min, the gas protector opened, and the flow rate was 20 mL min^−1^. The column temperature was initially at 80 °C (held for 1 min), increased to 150 °C at a rate of 30 °C min^−1^, then to 210 °C at a rate of 3 °C min^−1^, and finally increased to 290 °C at a rate of 10 °C min^−1^, holding for 12 min. The injection temperature was 260 °C and the injection volume with the splitless mode was 1 μL. The total running time was 43 min. The carrier gas was helium at 1.2 mL min^−1^ with constant flow. The collision gas was argon, and the pressure was in the range of 1.0 mTorr. The QqQ mass spectrometer was run at 70 eV EI in the selected reaction monitoring (SRM) mode. The specific MS/MS conditions are shown in Appendix A.

### 3.4. Preparation and Characterization of SBA-15-C_18_

SBA-15 and SBA-15-C_18_ were synthesized according to the method described by Casado, N. et al. [28].

Preparation of SBA-15: Pluronic 123 (3 g) was dissolved in the mixture of water (90 mL) and 2.0 M HCl solution (94 mL). TEOS (5.5 mL) was then added, and the resulting mixture was stirred vigorously at 40 °C for 4 h for the purposes of prehydrolysis. Next, the mixture was transferred into a polypropylene bottle and stirred at 90 °C for 24 h. The solid compound was obtained by filtration, washed with water, and dried for 12 h at room temperature. The template was removed from the obtained material by refluxing in 95% EtOH for 24 h. Finally, the material was filtered and washed several times with water and EtOH, then dried at 50 °C (Figure 1A).

Preparation of SBA-15-C_18_: SBA-15 was functionalized with chlorodimethyl -octadecylsilane. Prior to the reaction, SBA-15 was dehydrated at 150 °C in vacuum for 2 h. Then, chlorodimethyloctadecylsilane (5 mL) was dissolved in toluene (100 mL) and, after stirring well, SBA-15 (1 g) powder was added. The alkylation was completed after about 6 h. At the end of the reaction, the powder was filtered and rinsed with a large quantity of toluene solution to remove the chlorodimethyl -octadecylsilane, and then dried in a vacuum to obtain SBA-15-C_18_ powder (Figure 1A).

SBA-15-C_18_ was characterized by Fourier transform infrared spectroscopy (FTIR) and scanning electron microscopy (SEM) (Figure 1B,C).

### 3.5. Sample Extraction and Cleanup

An appropriate sample (10.0 ± 0.1 g) was weighed into a 50 mL centrifuge tube, and then 10 mL acetonitrile was added. Then 3 g NaCl was added, shaken vigorously for 1 min, and centrifuged at 3800 rpm for 5 min. The supernatant was used for the further m-PFC process.

The principle and purification process of the m-PFC syringe are shown in Figure 4A. The m-PFC syringe contains two polymethylene sieve plates and adsorption materials. The adsorbent materials used in this study contain (Figure 4B): 5 mg of MWCNTs, 5 mg of SBA-15-C_18_, and 150 mg of MgSO_4_ for the watermelon, melon, asparagus, lotus root, strawberry, and cucumber samples; 8 mg of MWCNTs, 10 mg of PSA, 8 mg of SBA-15-C_18_, and 150 mg of MgSO_4_ for the leek samples; 8 mg of MWCNTs, 5 mg of SBA-15-C_18_, and 150 mg of MgSO_4_ for the crown daisy and leaf lettuce samples. Next, 1 mL of the initial extract was introduced into the m-PFC syringe tube for purification. When the syringe piston was pushed, all of the extract filtered through the adsorbent at a rate of 1 drop s^−1^. At the same time, the purified extract was synchronously filtered through the 0.22μm filter membrane under the syringe, and finally introduced into the automatic sampling vial for GC-MS/MS analysis.

### 3.6. Method Performances

The validation process of the analytical method was performed using the following parameters: linearity, LOQs, accuracy, and precision. An external standard method was used for quantitative analysis. Linearity was studied using matrix-matched calibration by analyzing samples. Five recoveries and reproducibility tests were performed for each sample at one fortification level (10 μg kg^−1^). The LOQs of the proposed method were determined by the lowest accepted fortification level.

### 3.7. Intake Risk Assessment

Dietary intake risk assessment combines the data on food consumption with chemical concentrations in food, including the assessment of dietary intake risk for acute and chronic exposures (corresponding to short- and long-term exposures). Short-term exposure mainly refers to periods of up to 24 h, while long-term exposure mainly refers to the average daily exposure over a life cycle.

For a chronic intake risk assessment, international estimated daily intakes (IEDI) were calculated by Equation (1) and then compared with acceptable daily intakes (ADI).
(1)$$\mathrm{IEDI}=\sum \frac{\mathrm{Food}\;\mathrm{chemical}\;\mathrm{concetration}\times \mathrm{Fi}}{\mathrm{bw}}$$
where Fi is the average food consumption (kg) and bw is the average body weight (kg). When the IEDI is less than the ADI, then the risk is acceptable; if it is the opposite, then it poses an unacceptable risk.

For the acute exposure assessment, additional information is required on residual information in a single sample or a single unit of crop. The calculation of acute dietary exposure varies from case to case depending on the food commodity. The international estimated short-term intakes (IESTI) were calculated by Equations (2)–(4) and compared with the acute reference doses (ARfD).

Case 1. The residue in a composite sample (raw or processed) reflects the residue level in a meal-sized portion of the commodity (unit weight less than 25 g). Strawberries, leek, and crown daisy belong to Case 1 in this study.
(2)$$\mathrm{IESTI}=\frac{\mathrm{LP}\times \mathrm{HR}}{\mathrm{bw}}$$

LP is the largest portion consumed (97.5th percentile of eaters), kg/d. HR is the highest residue, mg/kg.

Case 2. The residue of a meal-sized portion might be higher than the composite residue (where the whole fruit or vegetable unit weight is above 25 g).

Case 2a. The unit edible weight of the commodity is less than the largest portion weight. Melon, lotus root, asparagus, leaf lettuce, and cucumber belong to Case 2a in this study.
(3)$$\mathrm{IESTI}=\frac{U\times \mathrm{HR}\times v+\left(\mathrm{LP}-U\right)\times \mathrm{HR}}{\mathrm{bw}}$$
where U is the unit weight of the edible portion and kg. v is the variability factor, which is applied to the composite residue to estimate the residue level in a high residue unit. Further, it is assigned a value of 3 according to WHO and JMPR.

Case 2b. The unit edible weight of the raw commodity exceeds the largest portion weight. The watermelon belongs to Case 2b in this study.
(4)$$\mathrm{IESTI}=\frac{\mathrm{LP}\times \mathrm{HR}\times v}{\mathrm{bw}}$$

When the IESTI is less than the ARfD, then the risk is acceptable; if it is the opposite, then it poses an unacceptable risk.

## 4. Conclusions

A novel m-PFC method based on SBA-15-C18 with GC-MS/MS was developed for a multi-residue analysis of 139 pesticides in agricultural produce. In 2020–2022, a total of 314 samples of different matrices, including watermelon, melon, asparagus, lotus root, strawberry, leek, crown daisy, leaf lettuce, and cucumber—which were collected from Shandong Province (China)—were analyzed. No pesticide residues were detected in 66.5% of the 341 samples, but 30.6% were at or below the MRLs, and 2.9% were above the MRLs. The residues of procymidone were found to most often exceed the Chinese MRLs (1.3% of the samples). Tebuconazole was found most frequently in 22.0% of the samples analyzed; most were from the leek and crown daisy samples. Multiple residues of pesticide (i.e., the residues of two or more pesticides) were found in 87 (75.2%) of the positive samples. The evaluation of consumer health risk connected with pesticide residues in agricultural products shows that the chronic and acute risk of pesticide residues are in an acceptable range, and they do not pose a serious health problem.

## Figures and Tables

**Figure 1 molecules-28-02480-f001:**
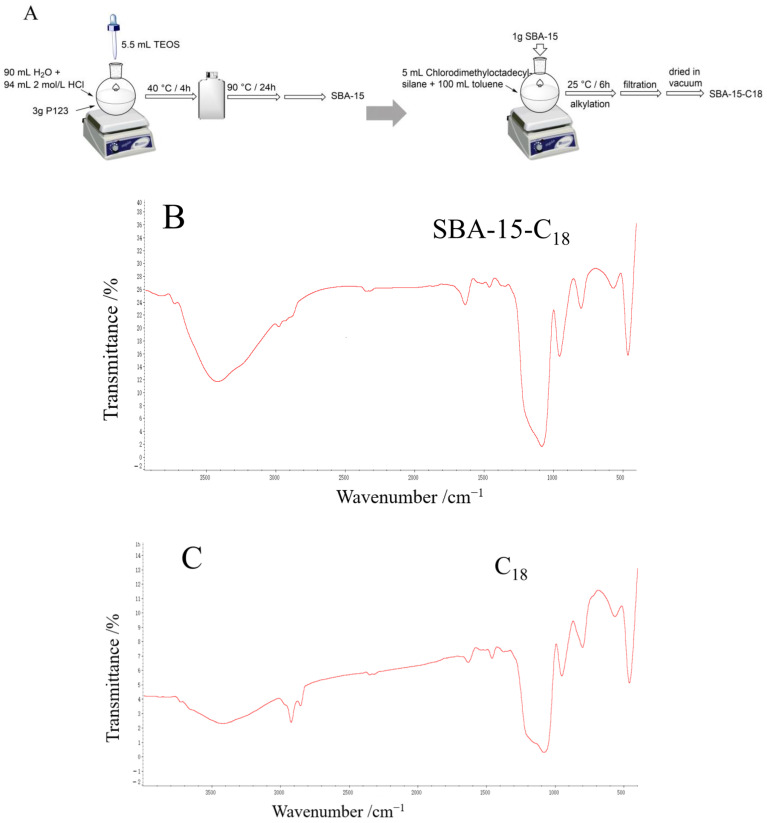
Synthesis and characterization of SBA-15-C_18_ and SBA-15. Synthesis diagram of SBA-15-C_18_ and SBA-15 (**A**). FTIR spectra of synthesized SBA-15-C_18_ (**B**) and C_18_ (**C**), as well as the SEM images of synthesized SBA-15-C_18_ (**D**) and SBA-15 (**E**).

**Figure 2 molecules-28-02480-f002:**
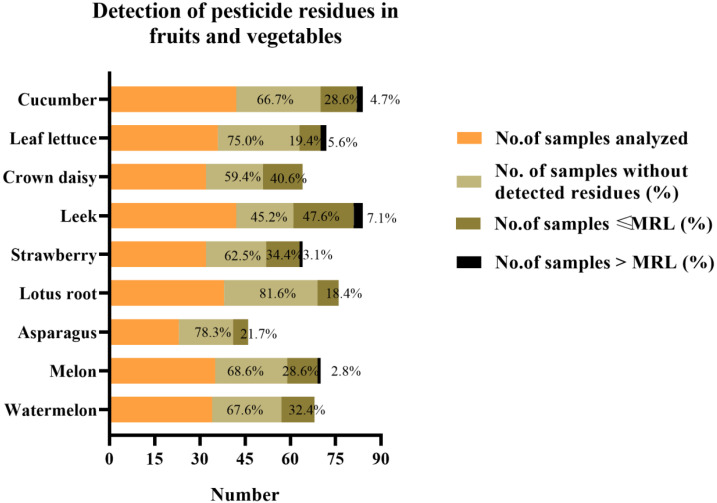
Frequency of samples with and without detected pesticide residues, as well as the samples containing residues above the MRLs for fruits and vegetables.

**Figure 3 molecules-28-02480-f003:**
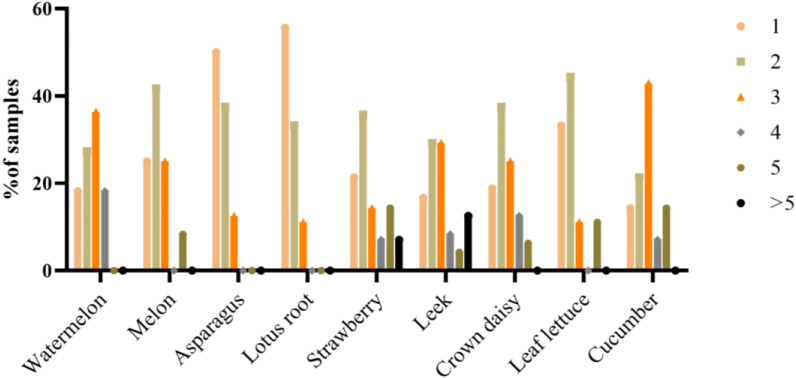
Commodities with multiple pesticide residues in the samples analyzed.

**Figure 4 molecules-28-02480-f004:**
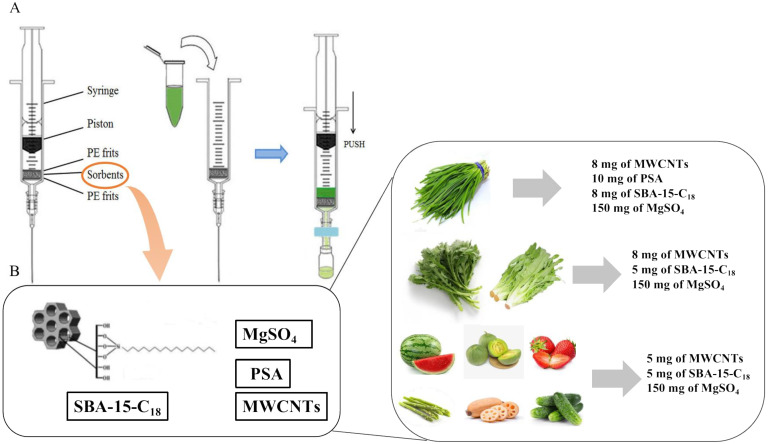
Schematic diagram of the m-PFC syringe and cleanup procedure (**A**), as well as the final sorbent combination and proportion for the different matrices (**B**).

**Table 1 molecules-28-02480-t001:** The standards and samples that exceed MRLs.

Samples	Triadimenol	Procymidone	Chlorpyrifos	Cypermethrin
	MRL/Detected Quantity (mg/kg)	MRL/Detected Quantity (mg/kg)	MRL/Detected Quantity (mg/kg)	MRL/Detected Quantity (mg/kg)
Melon	0.2/0.320	-	-	-
Strawberry	-	-	-	0.07/0.340
Leek1	-	0.2/11.270	-	-
Leek 2	-	0.2/10.560	-	-
Leek 3	-	0.2/5.430	-	-
Leaf lettuce 1	-	-	0.02/0.933	-
Leaf lettuce 2	-	-	-	2/2.344
Cucumber 1	-	2/2.334	-	-
Cucumber 2	0.2/0.510	-	-	-

**Table 2 molecules-28-02480-t002:** Results of the long-term intake and chronic exposure of pesticide residues.

Pesticide	ADI (mg/kg·bw·d)	Long-Term Risk		Pesticide	ADI (mg/kg·bw·d)	Long-Term Risk	
		IEDI (µg/kg·bw·d)	ADI%			IEDI (µg/kg·bw·d)	ADI%
Acetochlor	0.02	0.30	1.5	Flusilazole	0.007	0.00	0.0
Alachlor	0.01	0.01	0.1	Flutolanil	0.09	0.00	0.0
Ametryn	0.072	0.00	0.0	Kresoxim-methyl	0.4	0.02	0.0
Anilofos	0.001	0.02	2.2	Mefenacet	0.007	0.01	0.2
Atrazine	0.02	0.30	1.5	Mepronil	0.05	0.00	0.0
Azoxystrobin	0.2	0.01	0.0	Methidathion	0.001	0.00	0.0
Cypermethrin	0.02	0.32	1.6	Myclobutanil	0.03	0.01	0.0
Bifenthrin	0.01	0.01	0.1	DDT	0.01	0.01	0.1
Boscalid	0.04	0.00	0.0	Oxyfluorfen	0.03	0.14	0.5
Bromoxynil	0.01	0.00	0.0	Parathion	0.004	0.00	0.0
Butachlor	0.1	0.03	0.0	Pendimethalin	0.03	0.01	0.0
Butralin	0.2	0.00	0.0	Phenthoate	0.003	0.01	0.3
Chlorfenapyr	0.03	0.04	0.1	Pirimicarb	0.02	0.00	0.0
Chlorpropham	0.05	0.00	0.0	Probenazole	0.07	0.01	0.0
Chlorpyrifos	0.01	0.04	0.4	Procymidone	0.1	0.11	0.1
Clodinafop-propargyl	0.0003	0.17	5.5	Profenofos	0.03	0.09	0.3
Cyfluthrin	0.04	0.23	0.6	Prometryn	0.04	0.03	0.1
Cyhalothrin	0.02	0.07	0.3	Propachlor	0.54	0.29	0.1
Cyprodinil	0.03	0.00	0.0	Propargite	0.01	0.00	0.0
Deltamethrin	0.01	0.04	0.4	Propiconazole	0.07	0.02	0.0
Diclofop-methyl	0.0023	0.00	0.1	Diethofencarb	0.004	0.01	0.2
Difenzoquat	0.25	0.03	0.1	Pyriproxyfen	0.1	0.02	0.0
Endosulfan	0.006	0.04	0.6	Quinalphos	0.0005	0.01	1.4
Endrin	0.0002	0.02	9.5	Quizalofop	0.0009	0.01	0.6
Famoxadone	0.006	0.00	0.0	Spirodiclofen	0.01	2.42	24.2
Fenarimol	0.01	0.01	0.1	Tebuconazole	0.03	0.07	0.2
Fenbuconazole	0.03	0.01	0.0	Terbufos	0.0006	0.00	0.0
Fenobucarb	0.06	0.23	0.4	Tolfenpyrad	0.006	0.00	0.0
Fenpropathrin	0.03	0.03	0.1	Triadimefon	0.03	0.03	0.1
Fipronil	0.0002	0.00	0.0	Triadimenol	0.03	0.10	0.3
Fenthion	0.007	0.00	0.0	Triallate	0.025	0.02	0.1
Fluazifop-P-butyl	0.004	0.17	2.3	Trifloxystrobin	0.04	0.02	0.0
Fluroxypyr	1	0.03	0.0	Trifluralin	0.025	0.03	0.1

**Table 3 molecules-28-02480-t003:** Results of the short-term intake and acute exposure of pesticide residues.

Pesticide	ARfD (mg/kg·bw·d)	Commodity	Short-Term Risk		Pesticide	ARfD (mg/kg·bw·d)	Commodity	Short-Term Risk	
			IESTI (µg/kg·bw·d)	ARfD%				IESTI (µg/kg·bw·d)	ARfD%
Atrazine	0.1	Melon	0.37	0.4	Fenpropathrin	0.03	Leaf lettuce	0.44	1.5
		Watermelon	0.49	0.5	Fenpropathrin	0.03	Melon	0.25	0.8
		Asparagus	0.07	0.1	Fenthion	0.01	Leek	0.04	0.4
Cypermethrin	0.04	Strawberry	0.32	0.8			Asparagus	0.05	0.5
		Leek	0.62	1.6	Flusilazole	0.02	Melon	0.55	2.8
		Melon	5.23	13.1	Methidathion	0.01	Leaf lettuce	0.15	1.5
		Cucumber	1.04	2.6	Myclobutanil	0.3	Strawberry	8.57	2.9
		Leaf lettuce	0.53	1.3			Cucumber	0.30	0.1
Bifenthrin	0.01	Melon	1.22	12.2			Leaf lettuce	0.90	0.3
		Leaf lettuce	1.21	12.1	Pirimicarb	0.1	Asparagus	0.04	0.0
		Asparagus	0.19	1.9	Procymidone	0.1	Strawberry	2.90	2.9
Chlorfenapyr	0.03	Leek	14.76	49.2			Leek	58.30	58.3
Chlorpropham	0.5	Asparagus	0.05	0.0			Melon	13.56	13.6
Chlorpyrifos	0.1	Strawberry	0.37	0.4			Cucumber	2.67	2.7
		Leek	1.60	1.6			Watermelon	4.34	4.3
		Melon	8.09	8.1			Leaf lettuce	46.81	46.8
		Cucumber	0.45	0.5			Asparagus	0.08	0.1
		Leaf lettuce	0.17	0.2	Profenofos	1	Strawberry	5.24	0.5
		Asparagus	5.89	5.9			Cucumber	2.32	5.8
Cyfluthrin	0.04	Strawberry	0.32	0.8			Leaf lettuce	0.55	1.4
		Leek	0.59	1.5	Cyhalothrin	0.02	Strawberry	0.17	0.9
		Melon	1.65	4.1			Leek	0.08	0.4
		Melon	0.40	2.0			Watermelon	10.20	1.0
		Leaf lettuce	0.32	1.6	Propiconazole	0.3	Leaf lettuce	0.83	0.3
Deltamethrin	0.05	Strawberry	0.16	0.3	Tebuconazole	0.3	Strawberry	2.21	0.7
		Leek	0.09	0.2			Melon	0.88	0.3
		Melon	0.37	0.7			Cucumber	0.20	0.1
		Leaf lettuce	0.22	0.4			Watermelon	1.19	0.4
Endosulfan	0.02	Strawberry	3.80	19.0			Leaf lettuce	2.81	0.9
		Leek	0.06	0.3			Asparagus	10.11	3.4
		Melon	0.67	3.4					
		Cucumber	1.47	7.4	Tolfenpyrad	0.01	Cucumber	0.07	0.7
		Leaf lettuce	0.85	4.3	Triadimefon	0.08	Strawberry	0.08	0.1
Famoxadone	0.6	Leaf lettuce	0.50	0.1			Leek	0.08	0.1
buconazole	0.2	Cucumber	0.34	0.2			Leaf lettuce	0.44	0.6
		Watermelon	0.60	0.3	Triadimenol	0.08	Melon	9.83	12.3
		Melon	3.57	0.4			Watermelon	16.96	21.2

## Data Availability

Data is available on request from the corresponding author.

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
