# Peer review of "Residue Levels and Dietary Intake Risk Assessments of 139 Pesticides in Agricultural Produce Using the m-PFC Method Based on SBA-15-C_18_ with GC-MS/MS"

_molecules, 2023, doi:10.3390/molecules28062480_

Round 1

Author Response

Response to Reviewer 1 Comments

1.Please explain the material SBA-15-C18 in more detail in the “Introduction”, why this material was chosen and what were its advantages compared with C18?

Response 1: Thank you for the advices on the paper. More description of the SBA-15-C18 and its advantages over the C18 were added in “Introduction”. Please refer to line 64-70 on Page 2 in the revised manuscript.

2.Is there any reference on the synthesis process of SBA-15-C18? Is there any improvements the methods used in this paper compared with those used in the literature?

Response 2: We have added related reference on the synthesis process of SBA-15-C18. Please refer to line 136-137 on Page 5 in the revised manuscript. Compared with the methods used in the literatures, the improvement of our method lies in that, first, in the first step of preparing SBA-15, we raised the reaction temperature to 40 ºC, and the reaction time was reduced to 4 h, while the reaction proceeded at 35 ºC for 20 h in the literature. Second, in the literature, the unreacted templates were removed at 550 ºC by a temperature program, while the method in our paper was to remove the unreacted templates by refluxing in 95% EtOH, which is safer and avoids the danger brought by high temperature.

3.What are the main factors affecting the structure of SBA-15-C18?

Response 3: The structure of SBA-15 and SBA-15-C18 was compared by scanning electron microscopy, and it was found that the bonding of C18 groups had no significant effect on the structure of SBA-15-C18, and the structure of SBA-15-C18 was still two-dimensional through pore and regular pore structures.

4.The origin of the samples was not detailed. A total of 314 samples of 9 agricultural products were from which cities in Shandong Province? How to ensure the representativeness of these sample?

Response 4: Sample source and sample description were added in “2.2 Sample collection”. Please refer to line 108-109 on Page 4 in the revised manuscript. All samples were from local production, and the specific information of samples were shown in Table S2. The sampling was done according to guideline in China (Shandong) on sampling for official control of pesticide residues.

5.What was the source of the ADI?

Response 5: The source of the ADI was the Standardization Administration of China (GB2763-2021)

6.What case corresponds to each crop analyzed in dietary intake risk assessment?

Response 6: From IESTI Caculator, the case of each crop was: Watermelon: case 2b, melon: case 2a, asparagus: case 2a, lotus root: case 2a, strawberry: case 1, leek: case 1, crown daisy: case 1, leaf lettuce: case 2a and cucumber: case 2a.

7.please analysis the reasons, why procymidone and tebuconazole were found most often exceeding the ChineseMRLs .

Response 7: Thank you for the advices on the paper. The reason for procymidone and tebuconazole were found most often exceeding the ChineseMRLs has been added to the "Results and Discussion". Please refer to line 299-317 on Page 11 in the revised manuscript.

8.The purification methods and materials should be summarized in the "Conclusion".

Response 8: Thank you for the advices on the paper. The m-PFC method and SBA-15-C18 were added in the "Conclusion". Please refer to line 351-352 on Page 15 in the revised manuscript.

9.Please list the MRL values in Table 4

Response 9: Thank you for the advices on the paper. We have compiled a table of MRLs, listing MRL standards and samples that exceed MRLs. Please refer to Table 1 in the revised manuscript.

Reviewer 2 Report

The manuscript entitled “Residue levels and dietary intake risk assessments of 139 pesticides in agricultural produce using the m-PFC method based on SBA-15-C18 with GC-MS/MS” is interesting and well described. However, I believe that emphasis has been given mainly in the monitoring of pesticide residues and the evaluation of consumer safety and not on the analytical part and the detection of the targeted compounds. Therefore, the title is not considered appropriate.

Keywords: Please replace “risk assessment” with “dietary risk assessment”.

Introduction

Line 40-42: Please provide additional references as to support the specific statement.

Materials and Methods

“2.2. Sample collection”: I would recommend the authors to report the origin of the samples collected, e.g. domestic of import.

Line 157: It would be preferable to replace “A 1 mL” with “One mL”.

“2.6. Method Performances”: During the validation of an analytical method all related parameters much be examined. In the present work no reference is made in specificity and matrix effect assessment. Could the authors report any results on the above, if conducted? Additionally, the LOQ of the method is set at the lowest level in which acceptable accuracy and precision results were obtained. The criterion of S/N = 10 is also applied but usual as suuplementary.

The presentation of dietary intake risk assessment is too extended.

Results and Discussion

Line 230-238: The determination of residues with the m-PFC method based on SBA-15-C18 with GC-MS/MS has to be further discussed, providing its predominance against different analytical approaches. Comparison with recent similar literature should also be conducted.  

Line 239: “Matrix-matched standard calibrations were used for the more accurate results in or- 239 der to avoid matrix effect.” In which matrix were the calibration standards constructed? According to Table S1 only results in leek are reported.

The validation results should have been summarized and discussed in the main text of the manuscripts and not on be presented as supplementary material.

Line 255-262: The MRL values, well as the legal framework under which they were set, should be reported for each combination of commodity and analyte detected. Possibly in Table S4.

To my opinion chronic risk assessment cannot be applied after monitoring of single values of pesticide residues. The comparison of the obtained concentration with the respective ARfD seems more appropriate.

Author Response

Response to Reviewer 2 Comments

The manuscript entitled “Residue levels and dietary intake risk assessments of 139 pesticides in agricultural produce using the m-PFC method based on SBA-15-C18 with GC-MS/MS” is interesting and well described. However, I believe that emphasis has been given mainly in the monitoring of pesticide residues and the evaluation of consumer safety and not on the analytical part and the detection of the targeted compounds. Therefore, the title is not considered appropriate.

Response Pesticide residue and risk assessment is really the focus of this article. However, we believe that the purification method based on novel materials is an innovation point and highlight of this paper, so we hope to reflect the purification method and materials in the title.

Keywords: Please replace “risk assessment” with “dietary risk assessment”.

ResponseThank you for the advice on the paper. The key words have been revised as “Multiplug filtration cleanup; SBA-15-C18; Pesticide residue; Dietary risk assessment”.

Introduction:Line 40-42: Please provide additional references as to support the specific statement.

ResponseThank you for the advices. We have added related two reference published in food control in 2018 and 2021 that address the issue of pesticide residues in vegetables and fruits, supporting the specific statement. Please refer to “reference 5 and reference 6” in the revised manuscript.

Materials and Methods

“2.2. Sample collection”: I would recommend the authors to report the origin of the samples collected, e.g. domestic of import.

Response We have listed the sample sources in detail in Table S2 in the supplementary materials, including the cities and placessupermarket, market, or vegetable baseof the samples collected. Please refer to line 108-109 on Page 4 in the revised manuscript.

Line 157: It would be preferable to replace “A 1 mL” with “One mL”.

ResponseThis sentence has been revised. Please refer to line 171 on Page 6 in the revised manuscript.

“2.6. Method Performances”: During the validation of an analytical method all related parameters much be examined. In the present work no reference is made in specificity and matrix effect assessment. Could the authors report any results on the above, if conducted? Additionally, the LOQ of the method is set at the lowest level in which acceptable accuracy and precision results were obtained. The criterion of S/N = 10 is also applied but usual as suuplementary.

ResponseWe conducted all method validation for 139 pesticides in 9 agricultural products, including recoveries, repeatability, LOD, LOQ, correlation equation and R2. Due to the large amount of data, we listed the results of recoveries, LOQ and LOD of 9 kinds of agricultural products, as well as the correlation equation and R2 of leek samples in the supplementary materials (Table S3 and S4). For each pesticide, SRM mode was used in the QqQ mass spectrometer, and two pairs of ions were used to ensure the specificity of each pesticide in the detection. Matrix effects were evaluated and we found that the matrix effects of most pesticides couldn’t be ignored. Therefore, matrix-matched calibration was used for quantitative of pesticides for each matrix.

Regarding the setting of LOQ, we think your suggestion is very reasonable. However, for the analysis of multi-residue pesticides, it is difficult to set the lowest level of acceptable precision and precision results for each pesticide, so the standard of S/N = 10 is adopted. Thank you very much for your advice.

The presentation of dietary intake risk assessment is too extended.

ResponseThank you for the advices on the paper. We simplified and targeted the presentation of dietary intake risk assessment. Please refer to line 223-185 on Page 8 in the revised manuscript.

Results and Discussion

Line 230-238: The determination of residues with the m-PFC method based on SBA-15-C18 with GC-MS/MS has to be further discussed, providing its predominance against different analytical approaches. Comparison with recent similar literature should also be conducted.  

Response Thank you for the good advices. Our research group has published several previous papers on the application of m-PFC purification methods based on conventional MWCNTS and C18Zou, N. , et al. J. Agr. Food Chem. 2016, 64(31), 6061-6070. According to the results of this paper, SBA-15-C18 can achieve better purification effect with less dosage compared with C18. For example, for leek samples, the purification materials in the published literature were 8 mg MWCNTs + 10 mg PSA + 10 mg C18 + 150 mg MgSO4, which were 8 mg of MWCNTs+10 mg of PSA+ 8 mg of SBA-15-C18+ and 150 mg of MgSO4 in this paper. At the same time, our research group has published a literature in Chinese domestic journals (Song, Y., Shang-Ke, L., Jun-Jie, Z., Pei-Jie, H., Chang-Peng, C., & Nan, Z. Detection of 10 Kinds of Pesticide Residues in Tea Based on Novel Mesoporous Materials-QuEChERS-Ultra Performance Liquid Chromatography-Tandem Mass Spectrometry. CHINESE JOURNAL OF ANALYTICAL CHEMISTRY, 2021, 49(5), 827-835) to comprehensively evaluate SBA-15-C18 materials, and authorized a Chinese invention patent (Patent license number: CN 112516958 B). Therefore, the m-PFC method based on SBA-15-C18 has the advantages of excellent purification effect and less consumption.

Line 239: “Matrix-matched standard calibrations were used for the more accurate results in order to avoid matrix effect.” In which matrix were the calibration standards constructed? According to Table S1 only results in leek are reported.

ResponseWe constructed matrix-matched standard calibrations for each matrix, used to quantify pesticides in each matrix. Due to the large amount of data, we take leek as an example and only list the data of leek in Table S2.

The validation results should have been summarized and discussed in the main text of the manuscripts and not on be presented as supplementary material.

Response

The detailed data of method validation were presented in the supplementary material, and the validation results have been summarized in the text. Please refer to line 239-252 on Page 8 in the revised manuscript.

Line 255-262: The MRL values, well as the legal framework under which they were set, should be reported for each combination of commodity and analyte detected. Possibly in Table S4.

Response:Thank you for the good advices. We have compiled a table of MRLs, listing MRL standards and samples that exceed MRLs. Please refer to Table 1 in the revised manuscript.

To my opinion chronic risk assessment cannot be applied after monitoring of single values of pesticide residues. The comparison of the obtained concentration with the respective ARfD seems more appropriate.

ResponseSingle values of pesticide residues are not suitable for chronic risk assessment. However, in this study, we were not able to sample and analyze every food group in the human diet to make a very complete chronic risk assessment. Therefore, at present, IEDI was calculated based on single values of pesticide residues, the average food consumption and the average body weight, and compared with ADI to assess whether chronic risk is acceptable.

Round 2

Reviewer 2 Report

Authors ahave been taken into acount all the recommendations given and have revised their manuscript accordingly. Thank you. 

However, I strongly believe that the LOQ of the proposed method should reflect the lower accepted fortification level for each analyte. Otherwise, it is questionnable if the study can be reproduced successfully and additionally, its the applicability of the method in routine analysis. I suggest authors should present the LOQ of each compound based on the above criterion. 

Author Response

Response:Thank you for your advice. We think your suggestion is reasonable and correct. Therefore, we have modified the determination criteria to use the lowest acceptable fortification level as the LOQ. The LOQs for each compound were detailed in Table S3. Meanwhile, based on this criterion, we adjusted the detected values of pesticides, and detailed results were shown in Table S5. Please refer to line 173-174 on Page 5 and line 238-240 on Page 7 in the revised manuscript.